# Livestock and rodents within an endemic focus of Visceral Leishmaniasis are not reservoir hosts for *Leishmania donovani*

Anurag Kumar Kushwaha[1], Ashish Shukla[1], Breanna M. Scorza[2], Tulika Kumari Rai[1], Rahul Chaubey[3], Dharmendra Kumar Maurya[4], Shweta Srivastva[1], Shreya Upadhyay[1], Abhishek Kumar Singh[1], Paritosh Malviya[1], Om Prakash Singh[1,4], Vivek Kumar Scholar[3], Puja Tiwary[1], Shakti Kumar Singh[3], Phillip Lawyer[5], Edgar Rowton[6], Scott A. Bernhardt[7], Christine A. Petersen[2,8]*, Shyam Sundar[1]*

1 Department of Medicine, Institute of Medical Sciences, Banaras Hindu University, Varanasi, India, 2 Department of Epidemiology, College of Public Health, University of Iowa, Iowa City, Iowa, United States of America, 3 Kala-Azar Medical Research Center, Muzaffarpur, Bihar, India, 4 Department of Biochemistry, Institute of Science, Banaras Hindu University, Varanasi, India, 5 Arthropod Collections, Monte L. Bean Life Science Museum, Brigham Young University, Provo, Utah, United States of America, 6 Division of Entomology, Walter Reed Army Institute of Research, Silver Spring, Maryland, United States of America, 7 Department of Biology, Utah State University, Logan, Utah, United States of America, 8 Center for Emerging Infectious Diseases, University of Iowa, Coralville, Iowa, United States of America

* christine-petersen@uiowa.edu (CAP); drshyamsundar@hotmail.com (SS)

**Data Availability Statement:** All relevant data are within the manuscript.

## Abstract

Leishmaniasis on the Indian subcontinent is thought to have an anthroponotic transmission cycle. There is no direct evidence that a mammalian host other than humans can be infected with *Leishmania donovani* and transmit infection to the sand fly vector. The aim of the present study was to evaluate the impact of sand fly feeding on other domestic species and provide clinical evidence regarding possible non-human reservoirs through experimental sand fly feeding on cows, water buffalo goats and rodents. We performed xenodiagnosis using colonized *Phlebotomus argentipes* sand flies to feed on animals residing in villages with active *Leishmania* transmission based on current human cases. Xenodiagnoses on mammals within the endemic area were performed and blood-fed flies were analyzed for the presence of *Leishmania* via qPCR 48hrs after feeding. Blood samples were also collected from these mammals for qPCR and serology. Although we found evidence of *Leishmania* infection within some domestic mammals, they were not infectious to vector sand flies. Monitoring infection in sand flies and non-human blood meal sources in endemic villages leads to scientific proof of exposure and parasitemia in resident mammals. Lack of infectiousness of these domestic mammals to vector sand flies indicates that they likely play no role, or a very limited role in *Leishmania donovani* transmission to people in Bihar. Therefore, a surveillance system in the peri-/post-elimination phase of visceral leishmaniasis (VL) must monitor absence of transmission. Continued surveillance of domestic mammals in outbreak villages is necessary to ensure that a non-human reservoir is not established, including domestic mammals not present in this study, specifically dogs.

**Funding:** This work was supported by the Extramural Program of the National Institute of Allergy and Infectious Diseases, National Institutes of Health (NIH TMRC U19AI074321 to SS). The funders had no role in study design, data collection and analysis, decision to publish, or preparation of the manuscript.

**Competing interests:** The authors have declared that no competing interests exist.

## Author summary

*Leishmania donovani*, the causative agent of visceral leishmaniasis (VL), has unique enzootic, zoonotic and anthroponotic life cycles dependent on geographic region. Most *Leishmania* spp. are zoonotic, transmitted between humans and non-human mammals. Leishmaniasis is endemic in over 98 countries and the estimated population at risk on the Indian sub-continent (ISC) is around 0.2–0.4 million people. Herein, we assess knowledge gaps in disease transmission that challenge dogma regarding the reservoir(s) of visceral leishmaniasis and control efforts on the Indian sub-continent. Better understanding of *L. donovani* infection in domestic animals and its transmission to sand flies will provide answers to fundamental questions in VL epidemiology and ecology. Blood samples were collected from livestock and rodents in endemic villages and colonized sand flies were fed on these animals and analyzed post feeding to see if they could pick up parasites via bloodmeal. We found that livestock and to a smaller extent rodents could be infected with or exposed to *Leishmania* infection. However, these mammals were not infectious to *Phlebotomus argentipes*, the predominant vector species incriminated as transmitting *L. donovani* in India. The inability of these domestic mammals to transmit *L. donovani* to sand flies in Bihar suggests that they play a limited role in the spread of infection. A key component in reducing and preventing the re-emergence of VL in the context of the elimination program is to understand the role of vector feeding on non-human animals in the transmission cycle. Better estimation of the proportion of the livestock and rodent population exposed to sand flies during different seasons and ensuring that there are not reservoir species that contribute to transmission will help implement appropriate control strategies for sustainable elimination.

## 1.Introduction

Leishmaniasis is a vector-borne disease of humans and other mammals caused by at least 20 species of obligate protozoan parasites of the genus *Leishmania* (*Trypanosomatida*: *Trypanosomatidae*) with notable prevalence in more than 98 countries and territories [1]. Visceral leishmaniasis (VL), caused by *Leishmania donovani*- complex species, occurs in localized outbreaks within endemic regions of the Indian subcontinent, East Africa and Brazil with significant mortality [2–4]. *L. donovani* complex spp. infect humans as well as different mammalian species. Dependent on the presence or absence of a reservoir host, there are two primary types of epidemiological cycles: zoonotic VL, generally caused by *L. infantum*, with dogs as the primary reservoir host in the Mediterranean, the Middle East, Asia and South America; and anthroponotic VL, caused by *L. donovani* in the Indian subcontinent (ISC), with humans serving as the predominant mammalian host. VL is a major public health burden in the ISC, with 200 million residents at risk of infection, and roughly two thirds of the worldwide VL cases [2]. Over 61% of all Indian regions affected by VL (33 of 54 districts) are within Bihar state, reporting 70% of the VL burden from India [5]. Unlike in other regions of the world [6], VL is considered anthroponotic on the ISC due to the absence of direct evidence regarding a non-human reservoir [7,8]. In other regions, both humans and dogs are reservoirs unlike in India, or that are sources of infection to vectors [2].

Due to robust elimination efforts of the governments of India, Nepal and Bangladesh, as well as the World Health Organization, prevalence of VL on the ISC has reached its lowest level since the 1960s with an annual incidence of less than 1 per 10,000 population in 98%

(617/633) of Indian blocks by 2020 [9]. In this low-incidence setting, attention is focused on early screening in previously endemic areas and intervention in outbreaks. Outbreak investigations consist of surveillance procedures to establish an outbreak based on newly reported case numbers in the region then implementation of relevant interventions, and, when possible, identification of local risk factors to prevent parasite transmission [10]. To continue to have elimination-level case numbers over the coming years, it will be essential to maintain the ability of the public health system to forecast and monitor outbreaks of VL, as well as to quickly implement effective control measures. Indoor residual spraying (IRS) is used in endemic villages for vector management, impacting primarily sand flies resting inside dwellings. Amid mounting indications of resistance to dichlorodiphenyltrichloroethane (DDT), efforts across the ISC switched to use of synthetic pyrethroids in India by 2015 [5]. It has been hypothesized that due to the ecological pressure of IRS, remaining surviving sand flies have been selected for being less endophilic. *Phlebotomus argentipes* is the primary vector for *L. donovani* in this region, aggregating in animal shelters and mixed dwellings, which provide a suitable, steady micro-climate [11–13]. Sand flies have a proclivity for feeding on both ruminants and people in this habitat [12,14,15] and the potential to change infection dynamics by feeding on non-human hosts [16]. Studies in *L. donovani* foci on the ISC and Africa confirmed *Leishmania* exposure and infection in domestic mammals, but scientific evidence of their involvement as domestic reservoirs involved in transmission to people is still unclear [17–28]. Importantly, the genetically similar species, *Leishmania infantum*, found in Latin America and the Mediterranean basin is zoonotic with canids, including domestic dogs, foxes, jackals, lagomorphs, and wild rodents as significant reservoir hosts [29,30]. A high density of livestock is thought to be one of the risk factors for human VL in India [13]. There are also conflicting results regarding the role of domestic mammals as blood meal sources for *P. argentipes* and the resultant impact on VL epidemiology in the ISC [7]. For example, higher livestock density is associated with the reduction of *Leishmania* exposure in Nepal and Bangladesh [31,32], perhaps serving as an ecological sink for *L. donovani* infection, while increased VL risk is correlated with the presence of livestock in peri-domestic vegetation in India and Bangladesh [33–35].

Globally, epidemiological reports have identified evidence of infection or exposure to *L. donovani* in multiple domestic mammals indicates the possibility that *L. donovani* could be zoonotic [6]. However, to efficiently manage VL within a control program, it is necessary to determine the amount to which such non-human hosts are infectious to sand flies. Non-human sources of infection are not targeted through current VL elimination program interventions in the ISC. Mathematical modelling and observational studies posited that livestock as hosts for human *L. donovani* infection in the ISC could potentiate outbreaks or ongoing infection [14,17,36–38]. Finding a significant number of infected animals epidemiologically connected to human *L. donovani* exposure as well as demonstrating the infectiousness of domestic livestock to sand flies would establish the extent to which these animals contribute to VL ecology in endemic villages [6]. Xenodiagnosis is the most effective way to establish *Leishmania* infectiousness of a specific host to sand flies [39]. Our study aims to detect *Leishmania* infection and/or exposure in livestock and rodents from VL endemic villages of Muzaffarpur district in Bihar, India and identify their relative infectiousness to sand flies.

## 2. Methodology

### 2.1 Ethics statement

This work was conducted with ethical approval (Letter No- CAEC/Dean/2014/CAEC/615) obtained from Institutional Review Committees of Banaras Hindu University, Varanasi, India; and Kala-azar Medical Research Centre (KAMRC), Muzaffarpur, India and University of Iowa

Institutional Animal Care and Use Committee (IACUC) protocol number 9041721. In addition to the IACUC protocol, we asked approval from head of households to work with their animals and verbally assured the safety and comfort of these animals during sampling.

## 2.2 Selection of study area and epidemiological database

The study area included 15 villages across Muzaffarpur district (26.07˚N, 85.45˚E) in Bihar State, India where VL is highly endemic. The humid tropical temperatures of Muzaffarpur range from 14˚C in December–January to 32˚C in April–May with average annual precipitation of about 1,300 mm during the monsoon season from June to September. Villages with active transmission were selected based on current and past VL history from the Muzaffarpur-TMRC Health and Demographic Surveillance System (HDSS), ongoing since 2007 [40] and Kala-azar Management Information System (KAMIS) [10] (**Fig 1**).

## 2.3 Sampling procedures

Sample size was based on the assumption that the true proportion of infective animals would likely be very low (<1/100) which will allow us to show with 90% power that the proportion infective in the sampled population is less than 2%. Using an exact one-sided binomial test at the 2.5% level, a sample size of n = 200, we would have at least 90% power to find a proportion infective in less than 2% of the overall group of animals. These calculations were made via R, running simulations of outcomes for samples sizes between 100–400.

Adult livestock of both sexes were selected for blood sampling and xenodiagnosis with prior approval from the head of household. Cows, water buffaloes and goats were restrained

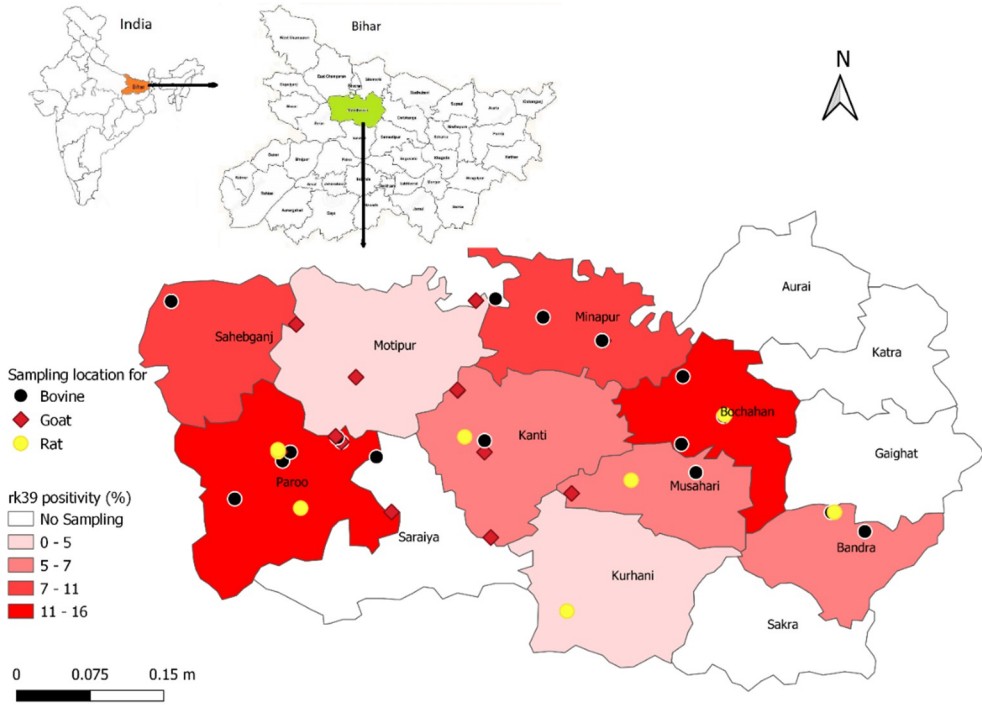

**Fig 1. Geographical location of livestock and rodents sampling and rk39 ELISA positivity across Muzaffarpur district Bihar, India.** Map base layers have adapted or reproduced from Malaviya et al., 2011 [70], published under CC.

via rope/halters and 1 ml whole blood was withdrawn from either the jugular or ear vein of each animal. Whole blood samples were collected in EDTA and red topped vacutainer tubes for serum and transported on ice packs (4˚C) from the field to the KAMRC laboratory where samples were aliquoted and stored at -20˚C for serological and molecular assays. Physical examination was performed on livestock, and history of any illness was noted. Rodents were caught using locally purchased live traps baited with bread. Traps were placed overnight in close proximity (~2 meters) to homes and livestock shelters. Fifty-one rats and 11 shrews and three voles were caught. Shrews were often found dead in the traps with no bread bait consumed. Voles and rats caught by live traps were immobilized in a plastic restraint and anaesthetized via intra-peritoneal injection of ketamine and xylazine according to body weight. All rodent blood was drawn after xenodiagnosis then euthanized via $CO_2$ chamber followed by splenic dissection. Spleen was kept in 1X PBS for subsequent DNA extraction at -20˚C.

## 2.4 Preparation of sand flies for xenodiagnosis

*Phlebotomus argentipes* sand flies from a closed, certified pathogen-free colony maintained at the KAMRC were used for xenodiagnosis [41]. Depending on the blood source and the proportion of flies in the feeding cup, feeding success ranged from 5 to 65 percent. With 30–50 females per feeding cup, close to optimal feeding (>60%) was achieved (plus 10 males). These findings and prior investigations led to a decision to limit the number of female *P. argentipes* per meal to 30–35, plus 10 males, and expose two feeding cups per mammals. The females used were three to five-day old (mature) and 12-hour starved [41,42].

## 2.5 Xenodiagnosis on animals

Infectiousness of livestock and rodents was tested by direct feeding of 30–35 female *P. argentipes* as previously described [42–46]. Feeding cups were placed on ears and axillary areas of livestock for 30 minutes under local environmental conditions (July- Monsoon, Feb- Winter, Oct- Spring) while xenodiagnosis on rodents were done in laboratory (**Fig 2**). Due to poor physical condition due to not consuming the bait overnight, shrews and voles were not used for xenodiagnosis. Once back to the insectary, blood-engorged females were placed into 1-pint paper cups and kept in an environmental chamber at 28˚C and 80% humidity for 48 hrs.

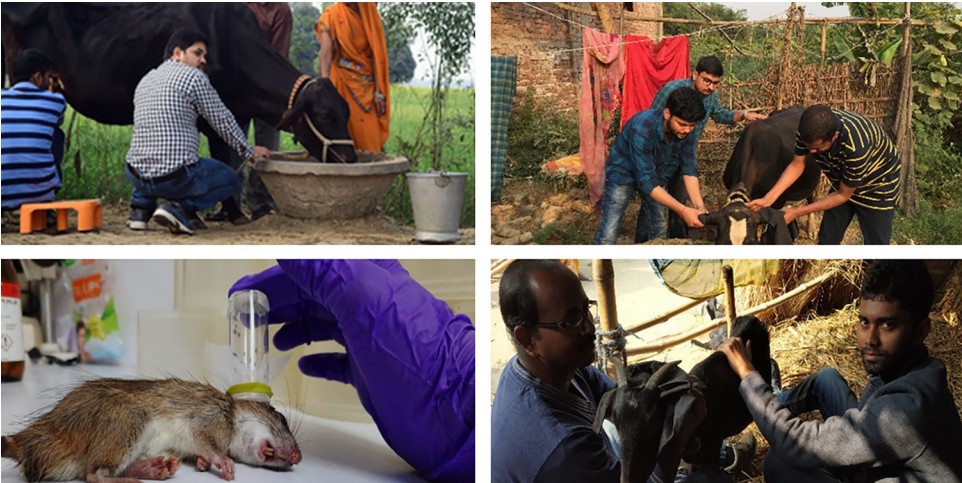

**Fig 2. Process of xenodiagnosis on different domestic animals in village settings.** Each person featured in the photo provided consent for the photo to be used in this article.

provided with 30% sugar-saturated cotton balls on the screen top. After 48 hrs. of an infected blood meal, the peritrophic membrane breaks down and procyclic promastigotes and necto-monads are detectable in sand fly midgut. Because of this after 48 hrs., blood fed flies were stored in 70% ethanol at -20˚C for further DNA extraction and qPCR for *L. donovani* [41]. The whole fly was processed for extraction.

## 2.6 DNA extraction from whole blood, splenic biopsy, and blood- fed sand flies

DNA was extracted from blood (200 μl) and spleen (25mg) by QIAamp Blood and tissue DNA mini kit (Qiagen, Hilden, Germany) in 60 μl of nuclease free water (Milli-Q) in accordance with the manufacturer's instructions. From single blood-fed sand flies, DNA was extracted 48 hrs. after xenodiagnosis via Gentra Puregene Tissue DNA Extraction Kit (Qiagen). To achieve maximum yield, tissue digestion was performed overnight with proteinase K in lysis buffer and increased the protein precipitation timing at -20˚C for overnight. DNA was eluted in 30 μl of Milli-Q and stored at -20˚C for downstream processing. This protocol was optimized for individual sand flies [46]. The quality of extracted DNA from samples (Blood, tissue and sand flies) was assessed by spectrophotometer (ND-2000 spectrophotometer; Thermo Scientific, USA). DNA samples having 260/280 ratio in range 1.8–2.0 with 260/230 ratio above 1.5 were used for qPCR experiments, which included approximately 95% of samples.

## 2.7 Quantitative Polymerase Chain Reaction (qPCR)

Quantification of parasites in whole blood, spleen and from blood-fed flies was performed by real-time Polymerase Chain Reaction (qPCR). TaqMan based qPCR on each DNA sample was run in triplicate on an Applied Biosystems (ABI) 7500 Real Time PCR system (Thermo Fisher Scientific, USA) to amplify a *L. donovani* kinetoplast minicircle DNA target with forward (kDNA4GGGTGCAGAAATCCCGTTCA), reverse primer (kDNA4 CCCGGCCCTATTTTA CACCA) and probe (ACCCCCAGTTTCCCGCCCCG) [47]. Nuclease free water (Thermo Fisher Scientific, USA) and blood DNA from Non-endemic healthy control (NEHC) and DNA from pooled uninfected laboratory reared *P. argentipes* were used as negative controls. Quantification of parasite equivalents in the test samples were calculated with a specific set of standard samples (DNA from healthy human blood and uninfected sand flies spiked with serial dilution of cultured *Leishmania* parasites) run in parallel to each set of test samples, as previously described [47,48]. qPCR cutoff Ct value for blood/tissue was ≤ 35. For sand flies, an approximate Ct >30 was considered negative [49].

## 2.8 Serology

For measurement of antibodies in serum against *L. donovani* rk-39 antigen, 25ng/well of rK39 antigen (Infectious Disease Research Institute (IDRI), Seattle, USA) was coated to 96-well flat-bottom microtiter plates in 100μL coating buffer (0.1 M carbonate-bicarbonate buffer, pH 9.6) and incubated overnight at 4˚C as described by elsewhere [42,50,51]. After two hours of incubation, the plates were blocked with 150μL blocking buffer (1% BSA in 0.05 M phosphate buffer) at 25˚C. One hundred microliter of serum samples (1:200 dilution) were added to the plates and incubated at 25˚C for 30 minutes. The plates were washed with phosphate-buffered saline (PBS) containing 0.1% Tween-20 (pH 7.4) and incubated with 100μL peroxidase-conjugated Goat Anti-Bovine IgG (Invitrogen, Cat No. A18751) (1:32000 dilutions) for cow/buffalo, Donkey Anti-Goat IgG (Invitrogen, Cat No. A15999) (1:2000 dilutions) for goat and Goat Anti-Rat IgG (Merck, Cat No. AP136P) (1:5000 dilutions) for rat at 25˚C for 30 minutes. Plates were then rinsed four times and incubated with 100μL TMB substrate for 5 minutes in the

dark. The reaction was stopped using 50 μL of 0.1 N H2SO4 and optical density (OD) was measured at 450 nm. Each sample was assayed in duplicate. Non-endemic region (no VL cases reported in the last 15 years) animals' serum was used as negative controls. rK39 Rapid Diagnostic Test (RDT) (InBios, Seattle, USA) were also done on all serum samples according to the manufacturer's instruction.

### 2.9 Statistical analysis

Cut-off values for seroprevalence were determined by adding two standard deviations to the mean optical density (OD) of the control sera from non-endemic areas. The prevalence of infection by *Leishmania* parasites was estimated based on the host species and the diagnostic method used. The total PCR positivity and seroprevalence were compared according to the animal group used in this study. Statistical analyses were performed using Graph Pad Prism software version 8 for all analyses.

## 3. Results

### 3.1 The low positivity for qPCR in livestock

Blood samples were collected from 255 livestock (85 cows, 66 buffalo and 39 goats) and rodents (51 rat, 11 shrew and 3 voles) in endemic villages. Antibodies against *L. donovani* rK39 were detected in 10.58% (27 out of 255) serum samples while *Leishmania* DNA was found in 2 buffalo and 1 goat: 1.17% (3 out of 255) of these mammals (**Table 1**). No any qPCR positive livestock were found to have seropositive. None of the splenic biopsies of rodents were found positive for *leishmania* kDNA qPCR. Anti-*Leishmania* antibodies were detected only in 2.35% cow (2/85) and 3.03% buffalo (2/66) out of total 255 mammals (1.56%) via rK39 RDT.

  None of the rk39 RDT positive mammals were found positive for qPCR or rk39 ELISA. Endemic villages goat sera had significantly higher (p = 0.0002) OD than non-endemic goat rk39 ELISA average OD, indicating demonstrable exposure to *L. donovani* rk39 antigen in this group of animals (**Fig 3**). No statistical difference was observed when comparing seropositive cattle average OD to other mammal group (goats and rodent) with lower seroreactivity to *L. donovani* antigen.

### 3.2 Livestock and rodents are not infectious to sand fly via xenodiagnosis

We conducted xenodiagnosis to evaluate host-to-vector transmissibility and determine host infectiousness in endemic villages between 2018 and 2021. A total of 255 animals were enrolled, 192 animals (78.4%) had xenodiagnosis performed (**Table 2**). Infection status was not known at the time of xenodiagnosis. In total, 4,993 laboratory reared female sand flies were exposed to these mammals. This included two buffalo and one goat that had qPCR detectable parasitemia, among 17 seropositive mammals (8.85%, 17/192). Despite this

**Table 1. *L. donovani* exposure and infection status for domestic and peri-domestic mammals as measured by rk39 ELISA and qPCR from serum and blood samples, respectively, from Muzaffarpur, Bihar, India.**

| Animals | rk39 ELISA (animal +/ total, n/%) | Blood qPCR (animal +/ total) |
|---|---|---|
| Cows | 10/85 (11.76%) | 0/85 (0%) |
| Buffalo | 8/66 (12.12%) | 2/66 (3.03%) |
| Goats | 4/39 (10.25%) | 1/39 (2.56%) |
| Rodents | 5/65 (7.69%) | 0/65 (0%) |
| Total | 27/255 (10.58%) | 3/255 (1.17%) |

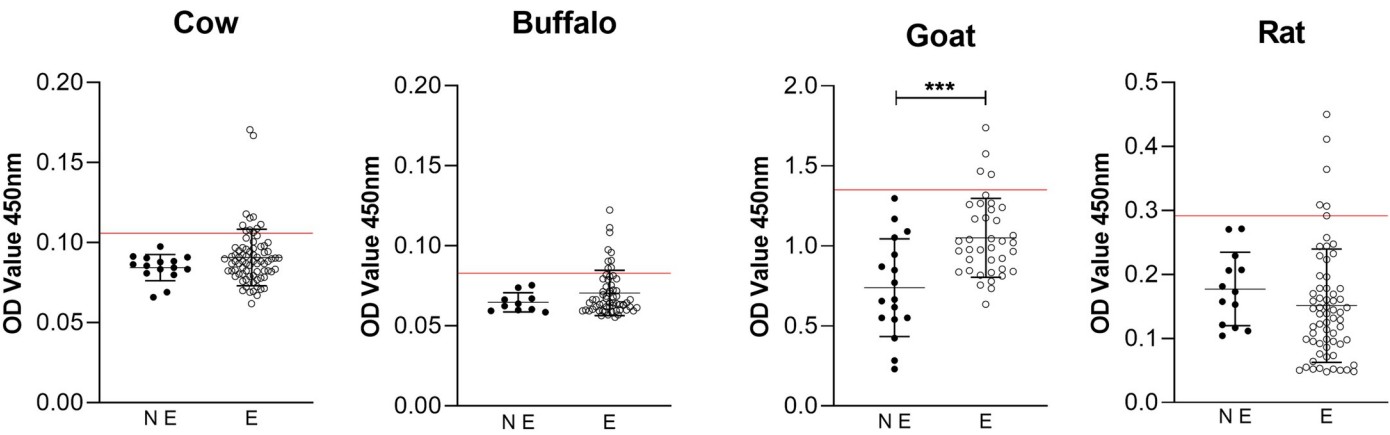

**Fig 3. Reactivity of cow, buffalo, goat and rodent sera from non-endemic locations (NE) and endemic villages (E) of Muzaffarpur for presence of anti *L. donovani* antibodies in rk39 ELISA.** Line indicates threshold to be considered seropositive based on mean OD of NE group + 2 standard deviations.

substantial sample size, no (0/4993) sand flies fed on these mammals were qPCR positive for *Leishmania* DNA.

## 4. Discussion

Many investigations have been carried out in East Africa and the ISC to find wild and/or domestic reservoir hosts of human VL. This study is the first investigation to confirm *Leishmania* infection occurring in livestock and rodents in a VL endemic area of India with the relationship of that infection to transmission to *P. argentipes* vector using xenodiagnosis. The host-targeted surveillance data, including tracking infection in sand flies and non-human favored blood meal sources of sand flies, and timeliness of responses to these data, are essential to monitor and sustain elimination by molecular and serological evidence of infection. The National Guidelines in India require a combination of human clinical symptoms and a positive serological and/or parasitological test to diagnose VL [5]. The most widely used diagnostic test in the ISC, the rK39 RDT, can only differentiate between active disease and asymptomatic infection in strict combination with clinical criteria [52] while qPCR positive or high rk39 ELISA titer shows very strong association to progression of VL [51]. Domestic livestock have also found to be *Leishmania* infected by other PCR and serological surveys in endemic villages of Bangladesh and Nepal at a higher prevalence than was found in Muzaffarpur by the current study [17,24]. *Leishmania* prevalence in different endemic areas is dependent on variable sand fly infection rates and distribution, which are influenced by numerous macro- and micro-environmental factors [53].

Even though there is active transmission of VL in the investigated region of Bihar, the epidemiology of infection in these mammals are still undocumented. To screen for the prevalence

**Table 2. Limited to no infectiousness observed from domestic animals via single blood-fed *P. argentipes* qPCR after xenodiagnosis.**

|  | # Animals | % Successful feeding | qPCR sand fly positive* animals +/tot (flies +/tot) |
|---|---|---|---|
| **Cows** | 47 | 47/47 (100%) | 0/47 (0/1636) |
| **Buffalo** | 41 | 41/41 (100%) | 0/41 (0/1674) |
| **Goats** | 39 | 39/39 (100%) | 0/39 (0/1108) |
| **Rodents** | 65 | 42/65 (65%) | 0/42 (0/575) |

of anti-*Leishmania* antibodies in livestock and rodents, we performed the ELISA and the rk39 RDT tests. Anti-*Leishmania* antibodies are a strong indication of exposure to infection by the parasite. A relatively high seroprevalence (11.92%, 18/151) in livestock (cow and buffalo) suggests this group is particularly exposed to *L. donovani*. Several livestock had a relatively strong antibody response, which could indicate sequential exposure to *Leishmania* parasites as a result of infected sand fly bites or that symptomatic infection occurred (Fig 1) [54]. Typically, these livestock (cows, buffaloes, and goats) are maintained in the communities and are usually in close proximity to the human inhabitants. *P. argentipes* prefers to feed on livestock and breeds primarily in cattle sheds. However, this vector also will feed on humans when in proximity [55–57]. This survey's finding of anti-*L donovani* antibodies in cow and buffalo serum, as detected by ELISA, is consistent with findings of prior studies conducted in Sudan [28], Bangladesh [24] and Nepal [17]. Anti-*Leishmania* antibodies were also discovered in domestic pigs in Brazil, however pigs were shown to be resistant to *Leishmania* infection, which may also occur in livestock explaining the discordance between seropositivity and PCR positivity rates [58]. It is also possible that the antibody response measured in livestock and rodents serum is due to cross-reacting antibodies inferred from other pathogens, which have been reported in human populations [59].

If parasites were unable to proliferate or survive within a given host, parasite DNA would be absent when using molecular diagnostic techniques. After peak transmission season, *Leishmania* parasites DNA were found in domestic mammals such as goats, cows, and buffaloes in Nepal [17]. We found evidence of *Leishmania* DNA only in two buffalo (2/66) and one goat (1/39). A positive PCR result likely indicated current or recent infection as DNA remains in the body for a short period of time [60] and infections may persist for several months during the highest peak of sand fly activity and *Leishmania* transmission season (May-June). It is believed that *L. donovani* is primarily transmitted between humans (anthroponotic), as opposed to genetically similar, zoonotic *L. infantum*, which is predominantly maintained in canine reservoirs [45,61]. Although this study detected *L. donovani* seropositivity in domestic mammals from endemic villages, these animals were not infectious to colony reared *P. argentipes* via xenodiagnosis. Detection of PCR-positive animals does not imply that these animal act as parasite reservoirs for sand flies. Instead, such species may act as parasite sinks, rather than contributing to the *Leishmania* infection cycle [62–64].

The normal *Leishmania* life cycle within the sand fly takes ~8–10 days to reach stationary phase growth [65]. It is possible that with better resources to extend the time post-xenodiagnosis for additional parasite replication to occur, *Leishmania* parasites may be detectable in sand flies fed on exposed livestock and rodents. However, qPCR is able to determine lower parasite loads and reliably quantify *Leishmania* from sand flies. Multiple works focused on evaluating parasite burden in sand flies used qPCR with primers targeting kDNA with high sensitivity and accuracy [61,66–68]. In general, absence or low quantity of parasites in peripheral blood or skin might explain failure of infectivity to sand flies. It has been reported that high parasite loads in the skin and blood of humans and dogs infected with *L. donovani* and *L. infantum* respectively were most correlated with infectiousness to sand fly vectors via xenodiagnosis [42,44,46,61,69]. As leishmaniasis is dermotrophic in nature it is important to evaluate skin parasite burden in transmission studies. The mammals included in this study (livestock) are culturally valuable, with cows specifically being sacred within Hindu religious practice in India. As such, it was not possible to get skin samples from livestock to evaluate parasite burden from these tissues.

The objective of this study was to assess the role of livestock and rodents in endemic areas for transmission of *L. donovani*, particularly considering the role of proximity and livestock density in altering human *Leishmania* incidence. Livestock are an important part of

households in the endemic settings on the ISC. Other animals, particularly dogs, have been demonstrated to be key reservoir species for the zoonotic transmission cycle of other visceralizing *Leishmania* parasites in other parts of the world. Additional studies are needed to establish the role of dogs in *L. donovani* transmission in endemic villages in Bihar, including xenodiagnosis. From this study and others, we conclude peri-domestic mammals, particularly livestock, were exposed to *L. donovani* parasites in endemic villages of Bihar. However, detectable parasitemia was rare. This leads us to conclude that there is a limited role for these mammals in transmission of *L. donovani* to humans. Given the propensity of sand flies to feed on domestic mammals, infection among livestock should continue to be monitored during outbreaks as sentinels and potential parasite sources.

## Acknowledgments

We acknowledge Mr. Anil Sharma and all the field staff at the KAMRC for administrative and logistical support. We also thank the friendly residents of the study villages and animal owners/caretakers without whose willing and outstanding cooperation this effort would have been impossible.

## Author Contributions

**Conceptualization:** Anurag Kumar Kushwaha, Christine A. Petersen, Shyam Sundar.

**Formal analysis:** Anurag Kumar Kushwaha, Breanna M. Scorza, Paritosh Malviya.

**Funding acquisition:** Christine A. Petersen, Shyam Sundar.

**Investigation:** Anurag Kumar Kushwaha, Ashish Shukla, Tulika Kumari Rai, Rahul Chaubey, Dharmendra Kumar Maurya, Shweta Srivastva, Shreya Upadhyay, Abhishek Kumar Singh, Vivek Kumar Scholar, Puja Tiwary, Shakti Kumar Singh, Phillip Lawyer, Edgar Rowton, Scott A. Bernhardt, Christine A. Petersen.

**Methodology:** Anurag Kumar Kushwaha, Ashish Shukla, Tulika Kumari Rai, Rahul Chaubey, Phillip Lawyer, Edgar Rowton, Scott A. Bernhardt, Christine A. Petersen.

**Project administration:** Om Prakash Singh, Vivek Kumar Scholar, Christine A. Petersen, Shyam Sundar.

**Resources:** Om Prakash Singh, Christine A. Petersen, Shyam Sundar.

**Supervision:** Christine A. Petersen, Shyam Sundar.

**Writing – original draft:** Anurag Kumar Kushwaha.

**Writing – review & editing:** Breanna M. Scorza, Om Prakash Singh, Phillip Lawyer, Scott A. Bernhardt, Christine A. Petersen.

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
