## [Decision Letter · Decision Letter 0]

10 May 2022

Dear Dr. Petersen,

Thank you very much for submitting your manuscript "Domestic Mammals as potential Reservoir Hosts for Leishmania donovani in India" for consideration at PLOS Neglected Tropical Diseases. As with all papers reviewed by the journal, your manuscript was reviewed by members of the editorial board and by several independent reviewers. In light of the reviews (below this email), we would like to invite the resubmission of a significantly-revised version that takes into account the reviewers' comments. 

Dear colleagues,

thank you for your submission. The manuscript was assessed by three expert reviewers who concur in there assessment that the paper, whilst of high interest to the field, needs a major revision along the lines suggested by the reviewers. I therefore ask you to revise the manuscript to meet the criteria and suggestions by the reviewers. Thank you in advance!

Joachim Clos

We cannot make any decision about publication until we have seen the revised manuscript and your response to the reviewers' comments. Your revised manuscript is also likely to be sent to reviewers for further evaluation.

Sincerely,

Joachim Clos

Associate Editor

Shan Lv

Deputy Editor

Dear colleagues,

thank you for your submission. The manuscript was assessed by three expert reviewers who concur in there assessment that the paper, whilst of high interest to the field, needs a major revision along the lines suggested by the reviewers. I therefore ask you to revise the manuscript to meet the criteria and suggestions by the reviewers. Thank you in advance!

Joachim Clos

Reviewer's Responses to Questions

**Key Review Criteria Required for Acceptance?**

**Methods**

-Are the objectives of the study clearly articulated with a clear testable hypothesis stated?

-Is the study design appropriate to address the stated objectives?

-Is the population clearly described and appropriate for the hypothesis being tested?

-Is the sample size sufficient to ensure adequate power to address the hypothesis being tested?

-Were correct statistical analysis used to support conclusions?

-Are there concerns about ethical or regulatory requirements being met?

Reviewer #1: Objectives are clearly stated.

The Study design does not allow testing infectiousness, and this is acknowledged by the authors in the discussion. 

The study population is clearly described, but not the rationale for the sample size. Therefore it is not possible to assess whether the selected sample size allows enough power to address the hypothesis. 

No concerns regarding ethical issues.

Reviewer #2: The authors should indicate in the methods how they decided on the number of sandflies for xenodiagnosis, and show that the number they used per animal species or group is sufficient to detect Leishmania infection, based on other xenodiagnosis studies.

See other and specific comments on the general comments.

Reviewer #3: (No Response)

**Results**

-Does the analysis presented match the analysis plan?

-Are the results clearly and completely presented?

-Are the figures (Tables, Images) of sufficient quality for clarity?

Reviewer #1: The results reflect the planned analysis and are clearly presented.

Reviewer #2: Results presented on tables are not mentioned in the text, and some results are only mentioned in the Discussion. 

See other and specific comments on the general comments.

Reviewer #3: (No Response)

**Conclusions**

-Are the conclusions supported by the data presented?

-Are the limitations of analysis clearly described?

-Do the authors discuss how these data can be helpful to advance our understanding of the topic under study?

-Is public health relevance addressed?

Reviewer #1: The conclusions reflect the data obtained. And some limitations are stated. It is missed a proposed plan for similar analyses/studies in the future.

Reviewer #2: Discussion of the limitations should be improved. The conclusions rest on the capacity of the xenodiagnosis to detect transmission with the sample size used, which needs to be demonstrated.

See other and specific comments on the general comments.

Reviewer #3: (No Response)

**Editorial and Data Presentation Modifications?**

Reviewer #1: Fine here.

Reviewer #2: See general comments.

Reviewer #3: (No Response)

**Summary and General Comments**

Reviewer #1: A great progress has been made on the Indian sub continent (ISC) to reach the targets set for visceral leishmaniasis (VL) in the WHO Road Map for NTDs, 2012-2020, i.e. Bangladesh, India and Nepal achieving elimination as a public health problem (<1 case/10 000 population, district or subdistrict level in endemic area). Pending validation, this target has been achieved by Bangladesh and Nepal, while India is very close (92%). In the new WHO Road Map for NTDs, 2021-2030 it is expected that the target will be reached by 2023. 

With this situation, activities like the one proposed by the authors are key in order to assess that no residual foci of transmission remain that can revert the current situation in case control efforts are relaxed. 

The study presented is pertinent and activities like this should be implemented across the ISC in order to gain understanding on the possibility of a zoonotic/anthropozoonotic cycle for Leishmania donovani.

There are however some points that in my opinión should be addressed/detailed in the manuscript in order to make this activity an example going forward and for others:

1) Xenodiagnosis, as it has been done, provides very little information. To assess infectivity to sand flies and the possibility of transmission of the parasites to another host, the screening for Leishmania parasites should have been conducted after blood meal digestion and defecation, which is beyond 48h. And ideally, infection should be confirmed by microscopy and observation of Leishmania parasites after dissecting the sandfly. 

2) Why the animals (livestock) were selected based on 'owner report of illness'? Why not including asymptomatic animals?

3) Selection of animals, what is the total (estimated) number of each of the species selected in the different subdistricts or sampling units? This figure provides the ecological context. Dogs were not included, despite being reservoirs for different Leishmania species, why?

4) Preparation of sand flies: what is the rationale for selecting 30-35 female Phlebotomus argentipes and no more or less? Authors could refer to other xenodiagnosis studies using 'hosts' for which infection has been confirmed. Even in these cases the infectivity is low. Therefore, are 35 sand flies enough to test infectiousness of the selected animals?

5) Statistical analysis. There is no mention to the rationale for selecting the number of animals and sandflies used in the study. 

6) Going forward, would the authors propose the same workflow? Would not be better to test for exposure (e.g. serology, PCR) and then conduct xenodiagnosis on seropositive or PCR positive animals?

7) Was it possible to get skin samples from the animals and test these for PCR? Would this be a better sample if one aims to assess their capacity as reservoirs?

Reviewer #2: 1. Lines 101-103: The authors should also refer to the L. donovani foci in Africa, as closely related to the ISC L. donovani, and the suspected reservoir hosts.

2. Section 1.1.: it would be interesting if the authors indicated the vector control measures in the studied areas, if any, and if known.

3. Line 144: what is an m-tube vial? Explain or use manufacturer name. 

4. Line 151: This should be in section 1.5. In any case, the “poor condition” should be explained. 

5. Line 152: Indicate how the rats were euthanized. 

6. Lines 160-161: Explain why this number of sandfly females per animal. Is this enough to detect infection? What are the references for this?

7. Line 162: Was the xenodiagnosis repeated throughout the year on rodents as well, or just livestock? If also on rodents, provide more detail.

8. Lines 163-164: Explain to the non-expert reader why female sandflies were kept for 48 hours post-feeding.

9. Line 166: Indicate at what temperature the samples were kept.

10. Section 1.7.: The reference for the qPCR (ref. 44?) should be given from the first sentence and, certainly, for the primers used.

11. Section 1.8.: A reference must be given to the rk39 ELISA, and it should be stated the origin or preparation of the rk39 antigen used. Furthermore, at the end of the section, it is stated that rK39 rapid tests were conducted, so the title of the section should be changed accordingly, and the authors should clarify if the ELISA and the rapid tests were done to all samples, and if not, to which samples and the selection algorithm. 

12. Lines 220-226: This belongs in the Introduction or the Discussion. 

13. Lines 225-226: Techniques, rather than technique. Indicate clearly if they are recommended together, or if only one is enough (“qPCR or rk39 ELISA”). In addition, qPCR is only meaningful together with the amplified target (or primers), so it should be indicated which qPCR is specifically recommended (with reference). 

14. Lines 226 and 228: “domestic animals” is not the best description for the sampled animals. It should be better “livestock and rodents”.

15. Line 228: Exposure is how the results are interpreted. Here, the authors should simply indicate the percentage of animals with a positive serology result. 

16. Lines 228-233: There is no need to repeat all the results presented on Table 1. Plus, the values are in Table 1 and not Figure 3, as indicated, which shows the distribution of OD values.

17. Line 239: Here should be the reference to Figure 3.

18. Section 2.1.: It should be mentioned in the text the low positivity results for qPCR. 

19. Figure 3, Legend: it should be indicated that the test was an ELISA and what NE and E means.

20. Lines 246-248: should be summarized to indicate that the authors have conducted xenodiagnosis to “quantify host-to-vector transmissibility and determining infectiousness of hosts” in endemic areas…

21. Line 260: species name in italics.

22. Lines 263-265: Both results should have been mentioned earlier in the Results.

23. Ensure consistency, throughout, regarding the use of RDT (Methods) and ICT dipstick test (Discussion). The same name or abbreviation should be used for the same test as from the first mention.

24. Also ensure consistency regarding the nomenclature for the animals: domestic animals, domestic cattle, livestock…

25. The Discussion should mention dogs, and their possible role or not in transmission in the ISC, as a complement to the animals surveyed in this work. 

26. The Discussion should compare these results with those obtained with other xenodiagnostic surveys of known Leishmania hosts, to evaluate how likely it would be to detect infected sandflies for each animal species surveyed, or demonstrate that the sample size for each animal species was adequate to detect infection.

27. The Discussion should be more explicit about the limitations of the work.

Reviewer #3: (No Response)

PLOS authors have the option to publish the peer review history of their article (what does this mean?). If published, this will include your full peer review and any attached files.

Reviewer #1: No

Reviewer #2: No

Reviewer #3: No
---

## [Decision Letter · Decision Letter 1]

11 Aug 2022

Dear Dr Petersen,

Thank you very much for submitting your manuscript "Livestock and rodents within an endemic focus of Visceral Leishmaniasis are not reservoir hosts for Leishmania donovani" for consideration at PLOS Neglected Tropical Diseases. As with all papers reviewed by the journal, your manuscript was reviewed by members of the editorial board and by several independent reviewers. The reviewers appreciated the attention to an important topic. Based on the reviews, we are likely to accept this manuscript for publication, providing that you modify the manuscript according to the review recommendations. 

Sincerely,

Joachim Clos

Academic Editor

Shan Lv

Section Editor

Dear colleagues,

I apologise for the delays in reaching a decision. With WorldLeish7 going on, I had problems securing a sufficient number of reviewers. I therefore have to go with the opinion of one reviewer and my own assessment. I ask you to address the points raised by Reviewer 2. In your response, please refer to page and line numbers where you addressed the questions and suggestions. 

I look forward to seeing the second revision.

Reviewer's Responses to Questions

**Key Review Criteria Required for Acceptance?**

**Methods**

-Are the objectives of the study clearly articulated with a clear testable hypothesis stated?

-Is the study design appropriate to address the stated objectives?

-Is the population clearly described and appropriate for the hypothesis being tested?

-Is the sample size sufficient to ensure adequate power to address the hypothesis being tested?

-Were correct statistical analysis used to support conclusions?

-Are there concerns about ethical or regulatory requirements being met?

Reviewer #2: See: Summary and General Comments

**Results**

-Does the analysis presented match the analysis plan?

-Are the results clearly and completely presented?

-Are the figures (Tables, Images) of sufficient quality for clarity?

Reviewer #2: See: Summary and General Comments

**Conclusions**

-Are the conclusions supported by the data presented?

-Are the limitations of analysis clearly described?

-Do the authors discuss how these data can be helpful to advance our understanding of the topic under study?

-Is public health relevance addressed?

Reviewer #2: See: Summary and General Comment

**Editorial and Data Presentation Modifications?**

Reviewer #2: Overall, there is too much repetition throughout the manuscript of information provided in previous sections. The manuscript should be cleaned up of this excess information.

Grammar issues (this is not an exhaustive list).

1. Line 39 – delete “but”

2. Line 55 – L. instead of Leishmania

3. Line 73 – add a comma after species.

4. Line 78 – add a comma after infantum.

5. Line 79 – add a semi-colon after America.

6. Line 81 – add “and” before “roughly”.

7. Lines 89 to 90 – This sentence requires a verb, or it should be added as a complement to the previous or the following sentence.

8. Line 95 – numbers rather than “counts”, which implies an action of counting.

9. Line 107 – “of” rather than “on”

10. Line 164 – “Depending” instead of “depends”

11. Line 212 – “elsewhere” instead of “by several findings”

12. Line 213 – Sentences should not start by a numeral, so 100 should be “One hundred…”

13. Line 236 – “Leishmania”.

**Summary and General Comments**

Reviewer #2: The authors have addressed many issues by the reviewers, but not all were completely integrated in the revised manuscript, or I was not able to identify where. Ensure that in the new response to reviewers includes the new or revised text and where the previous and new issues were addressed in the manuscript. 

Major issues

1. Despite the large number of sandflies used for xenodiagnosis, only 3 mammals had detectable Leishmania DNA, and it is not clear in the manuscript if these animals were included in the xenodiagnosis. Even if all seropositive mammals were infected, it is not given the percentage of the 192 animals that were seropositive, and it would be only 14% of all animals. 

2. The authors do not mention the estimated sensitivity of xenodiagnosis, and they should, but it would make more sense to concentrate the xenodiagnosis effort on animals with more likely L. donovani infection (high rk39 titre or positive qPCR), in terms of number of sandflies, but to conduct xenodiagnosis at different times during the day or along a week, for example. It is understandable if the authors have done the xenodiagnosis without prior knowledge of the infection status of the animals, but this should be discussed as a limitation of this work.

3. The authors provide quite a lot of the information asked by reviewers in reply to reviewers, but not enough in the manuscript for many issues, including selection of mammals for xenodiagnosis, or the use of skin samples, which should be included in the discussion. Review throughout questions and answers to reviewers. 

4. Given that the authors have first conducted xenodiagnosis and only after detection of infection, the Results should reflect this order, including data that shows an estimation of how many animals subjected to xenodiagnosis were likely infected. 

5. In my opinion, the authors have not addressed the question by all three reviewers regarding dogs as possible reservoirs of L. donovani in the ISC (or not), considering their role elsewhere, including Sudan, where L. donovani is also present, and the reasons why dogs were not included here. 

Minor issues:

1. Line 61 – indicate that Ph. argentipes is the vector of L. donovani in India, to explain why this species was used, either here or elsewhere in the Author Summary.

2. Line 66 – indicate which population. Domestic animals? 

3. Lines 74 to 75 – “…and notable prevalence of Leishmania in more than 98 countries and territories” – a verb seems to be missing and this part of the sentence doesn’t follow well from the initial part, which refers to L. donovani, but the 98 countries are for any species that cause leishmaniasis in humans. This should rather follow or add to the first sentence of the Introduction. 

4. Line 86 – Given that leishmaniasis is a vector-borne disease, the mammal host isn’t considered infectious. It is better to use “reservoir” here. The expression “that promote transmission” is not accurate either, as vectors actively seek to feed on mammals (infected or not). Finally, “to others” is not clear enough. Simply state that in other regions both humans and dogs are reservoirs unlike in India, or that are sources of infection to vectors. 

5. Line 92 – define what “normal” is intended to mean in this sentence, or replace with a more adequate word. Does it refer to number of cases in relation to those recorded in previous years? Or does it refer to the actual number of cases during the outbreak?

6. Lines 147 to 148 – Show calculations or indicate calculator used in the manuscript.

7. Line 155 – indicate model and manufacturer of traps.

8. Line 156 – define proximity in distance range. 

9. Lines 157 to 161 – it is only mentioned that blood was taken from rats, and that they were euthanized. Indicate what happened to the shrews and the voles here, rather than in section 1.5. (lines 174-176), for clarity. 

10. Lines 166 to 169 – This sentence is somewhat confusing. I think the issue is the placement of commas to indicate separate sections. I suggest: “… to limit the number of female P. argentipes per meal to 30-35, plus 10 males, and expose two feeding cups (…). The females used were three to five-day old (mature) and 12-hour starved.”

11. Lines 171 to 172, which indicate 30-35 flies, seem to contradict lines 166 to 169, which state 60-70 flies.

12. Line 181 – “ethanol” rather than “alcohol”, unless another alcohol was used, but then specify which.

13. Line 184 – indicate the weight or volume used of the biological material for DNA extraction.

14. Lines 186 to 187 – What is this precipitation time? As far as I’m aware there is no precipitation step using the QIAmp kits, unless to determine DNA length. If that was the case, explain that this was done, but also indicate what was the time used, not just “increased”. 

15. Section 1.6. Indicate if the elution volume and eluent was the same for both kits, and if the quality assessment was also done for both methods of DNA extraction. If not, ensure that the method description is equally complete for both methods.

16. Line 209 – add “L. donovani” before rk-39.

17. Section 1.8 – Volumes for all ELISA components should be given, not just of the sample serum.

18. Line 235 – indicate how many of each livestock and rodents.

19. Line 237 – indicate in which groups of mammals Leishmania DNA was found.

20. Line 237 – indicate if qPCR positive animals were also rk39 ELISA positive or not.

21. Line 239 – indicate %s for each group as well.

22. Lines 243 to 250 – indicate that the comparison is between rk39 ELISA average OD, presumably, or what was compared. Also provide the values under comparison.

23. Figure 3 – it would make more sense to have the groups of animals in the same order as Table 1.

24. Lines 250 to 252 – Please clarify is sentence. Is the comparison between seropositive and seronegative cattle? Or are “other animals” other groups? In any case, it’s not clear what is being compared. Is it average OD?

25. Lines 256 – how were the 192 animals selected? What were the criteria used? – it should be explained in Materials and Methods. Were the three mammals with detectable Leishmania DNA included in the xenodiagnosis?

PLOS authors have the option to publish the peer review history of their article (what does this mean?). If published, this will include your full peer review and any attached files.

Reviewer #2: No

Figure Files:

Data Requirements:

Reproducibility:

References

---

## [Decision Letter · Decision Letter 2]

12 Oct 2022

Dear Dr. Petersen,

We are pleased to inform you that your manuscript 'Livestock and rodents within an endemic focus of Visceral Leishmaniasis are not reservoir hosts for Leishmania donovani' has been provisionally accepted for publication in PLOS Neglected Tropical Diseases.

We would like to ask that you please correct the following sentence in the introduction, on page 4: "In addition, a zoonotic cycle of L. tropica transmitted by P. (Par) sergenti might also be present" to ". . . a zoonotic cycle of L. tropica transmitted by P. (Par) sergenti is also present," as there is no doubt about this and there are zoonotic cycles that are the primary mode of parasite circulation in several foci in the Middle East.

Before your manuscript can be formally accepted you will also need to complete some formatting changes, which you will receive in a follow up email. A member of our team will be in touch with a set of requests.

Best regards,

Shan Lv, Ph.D.

Section Editor

Reviewer's Responses to Questions

**Key Review Criteria Required for Acceptance?**

**Methods**

-Are the objectives of the study clearly articulated with a clear testable hypothesis stated?

-Is the study design appropriate to address the stated objectives?

-Is the population clearly described and appropriate for the hypothesis being tested?

-Is the sample size sufficient to ensure adequate power to address the hypothesis being tested?

-Were correct statistical analysis used to support conclusions?

-Are there concerns about ethical or regulatory requirements being met?

Reviewer #2: (No Response)

**Results**

-Does the analysis presented match the analysis plan?

-Are the results clearly and completely presented?

-Are the figures (Tables, Images) of sufficient quality for clarity?

Reviewer #2: (No Response)

**Conclusions**

-Are the conclusions supported by the data presented?

-Are the limitations of analysis clearly described?

-Do the authors discuss how these data can be helpful to advance our understanding of the topic under study?

-Is public health relevance addressed?

Reviewer #2: (No Response)

**Editorial and Data Presentation Modifications?**

Reviewer #2: (No Response)

**Summary and General Comments**

Reviewer #2: 1. Most of the reply to point 3 of "Major points" should be included in the Discussion and indicated in the reply where it was inserted.

2. Part of the reply to point 5 of "Major points" should be included in the Discussion and indicated in the reply where it was inserted. At the very least, indicate that dogs can be suspected reservoirs, as for other VL agents, and that they are the subject of a separate study. It should be clear to the reader that the omission of dogs from this work was not an oversight.

PLOS authors have the option to publish the peer review history of their article (what does this mean?). If published, this will include your full peer review and any attached files.

Reviewer #2: No

---

## [Editor Report · Acceptance letter]

16 Oct 2022

Dear Dr. Petersen,

We are delighted to inform you that your manuscript, "Livestock and rodents within an endemic focus of Visceral Leishmaniasis are not reservoir hosts for *Leishmania donovani*," has been formally accepted for publication in PLOS Neglected Tropical Diseases.

Best regards,

Shaden Kamhawi

co-Editor-in-Chief

Paul Brindley

co-Editor-in-Chief
